# Structures of β1-adrenergic receptor in complex with Gs and ligands of different efficacies

Minfei Su[1,5], Navid Paknejad [2,5], Lan Zhu[3], Jinan Wang[4], Hung Nguyen Do[4], Yinglong Miao [4], Wei Liu [3], Richard K. Hite [2✉] & Xin-Yun Huang [1✉]

G-protein-coupled receptors (GPCRs) receive signals from ligands with different efficacies, and transduce to heterotrimeric G-proteins to generate different degrees of physiological responses. Previous studies revealed how ligands with different efficacies activate GPCRs. Here, we investigate how a GPCR activates G-proteins upon binding ligands with different efficacies. We report the cryo-EM structures of β1-adrenergic receptor (β1-AR) in complex with Gs ($G\alpha_s G\beta_1 G\gamma_2$) and a partial agonist or a very weak partial agonist, and compare them to the β1-AR–Gs structure in complex with a full agonist. Analyses reveal similar overall complex architecture, with local conformational differences. Cellular functional studies with mutations of β1-AR residues show effects on the cellular signaling from β1-AR to the cAMP response initiated by the three different ligands, with residue-specific functional differences. Biochemical investigations uncover that the intermediate state complex comprising β1-AR and nucleotide-free Gs is more stable when binding a full agonist than a partial agonist. Molecular dynamics simulations support the local conformational flexibilities and different stabilities among the three complexes. These data provide insights into the ligand efficacy in the activation of GPCRs and G-proteins.

[1] Department of Physiology and Biophysics, Weill Cornell Medical College of Cornell University, New York, NY 10065, USA. [2] Structural Biology Program, Memorial Sloan Kettering Cancer Center, New York, NY 10065, USA. [3] Cancer Center and Department of Pharmacology and Toxicology, Medical College of Wisconsin, Milwaukee, WI 53226, USA. [4] Center for Computational Biology and Department of Molecular Biosciences, University of Kansas, Lawrence, KS 66047, USA. [5] These authors contributed equally: Minfei Su, Navid Paknejad. ✉email: hiter@mskcc.org; xyhuang@med.cornell.edu

G-protein-coupled receptors (GPCRs) mediate transmembrane signaling from ligands with different efficacies to downstream heterotrimeric G-proteins[1–5]. Ligands can vary in efficacy, namely in their intrinsic ability to activate downstream signaling pathways. Full agonists elicit the maximal signaling response, and partial agonists induce various degrees of sub-maximal responses. Antagonists produce no or little responses by themselves, but block the binding of other ligands to GPCRs. Inverse agonists decrease the basal physiological activity of GPCRs[1,5,6]. A conformation selection model has been proposed to explain the actions of ligands with different efficacies on GPCRs[4,5,7]. GPCRs are highly dynamic proteins and can sample multiple conformations including inactive states, intermediate states, and active states. These different conformations are in equilibrium with each other. Ligands stabilize unique and ligand-specific GPCR conformations. Binding of full agonists stabilizes the active state conformation, and shifts the GPCR conformational equilibrium towards the active states. Partial agonists select a different conformation and are less able to drive the equilibrium to the active state than full agonists. Therefore, the population or amount of GPCRs in the active state is correlated with ligand efficacy[4,5,7].

After ligand binding, GPCRs activate G-proteins to initiate downstream physiological responses. Here we have investigated the activation of G-proteins by GPCRs bound with ligands with different efficacies. Previous X-ray crystal structural studies of GPCRs (without G-proteins) bound with ligands with different efficacies surprisingly showed similar conformations for individual GPCRs that most are in the inactive states[4]. The ligand-binding pockets in the receptors adopt ligand-specific configurations. On the other hand, nuclear magnetic resonance and fluorescence life-time spectroscopy studies of GPCRs (without G-proteins) indicate that ligand efficacy correlates with local conformational changes, and these changes occur in a fast timescale[8–17]. Furthermore, X-ray crystal and cryo-EM structures of the complexes of full agonist-bound GPCRs and G-proteins show that GPCRs in these complexes are in the fully active states[18,19]. While the interactions between GPCRs (without G-proteins) and full agonists, partial agonists and antagonists have been investigated, we still do not fully understand the structural and biochemical bases for the activation of G-proteins by GPCRs after bound with partial agonists[18,19].

In this work, we use structural, computational, cellular and biochemical approaches to understand the mechanisms of activation of G-proteins by $\beta_1$-adrenergic receptor ($\beta_1$-AR) after bound with ligands of different efficacies. We determine the cryo-EM structures of $\beta_1$-AR and heterotrimeric Gs (G$\alpha_s$G$\beta_1$G$\gamma_2$) in complex with a partial agonist (dobutamine), or a very weak partial agonist (cyanopindolol; also called an antagonist). We then compare these cryo-EM structures with our previously determined cryo-EM structure of $\beta_1$-AR and Gs in complex with a full agonist (isoproterenol)[20]. This provides the opportunity to compare and contrast the interactions between the same GPCR and the G-protein in the presence of a full agonist, a partial agonist, and a very weak partial agonist. The comparison surprisingly reveals that the overall structures of the three different complexes are similar, with local conformational differences mainly in the ligand-binding pockets. Furthermore, we examine the activation of Gs by $\beta_1$-AR in cells after stimulation by these three ligands. We generate mutations in the residues on $\beta_1$-AR that interact with the ligands or Gs. These mutations impair the cellular signaling from $\beta_1$-AR to the downstream cAMP response initiated by the three different ligands, with residue-specific differences. Moreover, we investigate the stability of the intermediate state complex of the nucleotide-exchange process (i.e. the ligand–$\beta_1$-AR–nucleotide-free Gs complex). We find that, when

bound with a full agonist, the $\beta_1$-AR–Gs (nucleotide-free) intermediate state is more stable than the intermediate states bound with a partial agonist. All-atom simulations using a robust Gaussian accelerated molecular dynamics (GaMD) method[21,22] support these structural and biochemical findings. Together, these data provide insights into the activation of G-proteins by GPCRs after bound with ligands of different efficacies.

## Results

**Cryo-EM structures of the complexes of Gs and $\beta_1$-AR bound with a partial agonist or a very weak partial agonist.** To understand the activation of G-proteins by a GPCR bound with a partial agonist or a very weak partial agonist, we started with the structural studies. We used isoproterenol as an example of a full agonist, dobutamine as a partial agonist, and cyanopindolol as a very weak partial agonist for turkey $\beta_1$-AR[23] (Fig. 1a). We have solved a 2.6 Å cryo-EM structure of dobutamine-bound $\beta_1$-AR and Gs complex (Fig. 1b, Supplementary Fig. 1, and Supplementary Table 1), and a 2.5 Å cryo-EM structure of $\beta_1$-AR–Gs in complex with cyanopindolol (Fig. 1c, Supplementary Fig. 2, and Supplementary Table 2). We then compare and contrast these structures with the previously determined 2.6 Å cryo-EM structure of isoproterenol-bound $\beta_1$-AR and Gs complex[20]. The well-defined density maps allowed us to build structures of $\beta_1$-AR–Gs in the presence of an agonist, a partial agonist and a very weak partial agonist (Supplementary Fig. 3). Overall, the structures of the $\beta_1$-AR–Gs complex in the presence of isoproterenol, dobutamine, or cyanopindolol are similar. However, there are local conformational differences, especially in the ligand-binding pockets (Fig. 2, Supplementary Figs. 4-6). Rearrangements of critical ligand-binding residues (such as Phe201$^{ECL2}$) can be detected when comparing the isoproterenol and dobutamine-bound structures with the cyanopindolol-bound structure (Fig. 2a). While some of the interacting residues are common to all three ligands, including Trp117$^{3.28}$, Thr118$^{3.29}$, Asp121$^{3.32}$, Val122$^{3.33}$, Val125$^{3.36}$, Phe201$^{ECL2}$, Ser211$^{5.42}$, Ser215$^{5.46}$, Phe306$^{6.51}$, Asn310$^{6.55}$, Asn329$^{7.39}$, and Tyr333$^{7.43}$ (the superscript denotes the Ballestero-Weinstein numbering system)[24], dobutamine and cyanopindolol each make a unique set of additional interactions in the orthosteric ligand-binding pocket (Fig. 2b–d, Supplementary Fig. 7). Dobutamine is additionally coordinated by the backbone carbonyl oxygen of Gly98$^{2.61}$, and side chains of Leu101$^{2.64}$, Val102$^{2.65}$, Phe307$^{6.52}$, Val326$^{7.36}$, and Trp330$^{7.40}$ (Fig. 2c, Supplementary Fig. 7). Cyanopindolol, on the other hand, makes additional interactions with Thr126$^{3.37}$, Thr203$^{ECL2}$, Ala208$^{5.39}$, and Phe307$^{6.52}$ on the opposing side of the orthosteric ligand-binding pocket (Fig. 2d, Supplementary Fig. 7). These shared and distinct interactions are essential for the accommodation of the three ligands with different chemical scaffolds, and are similar to those observed in the complexes of these ligands with $\beta_1$-AR in the presence of a conformation-specific nanobody[25] (Fig. 2, Supplementary Figs. 3, 7, 8).

To functionally validate the structurally identified residues, we have mutated some shared and unique residues involved in ligand interactions (Fig. 3a–f). We mutated residues Leu101$^{2.64}$ (for Dob), Trp117$^{3.28}$ (for all three ligands), Thr203$^{ECL2}$ (for Cya), Val326$^{7.36}$ (for Dob), and Trp330$^{7.40}$ (for Dob) to Ala. These mutants were then expressed in cells, and their responses to different concentrations of the three different ligands (isoproterenol, dobutamine, and cyanopindolol) to generate cellular cAMPs were quantified (Fig. 3a–f, Supplementary Fig. 9). While Trp117Ala mutation decreased the potency and efficacy of all three different ligands, Leu101Ala, Val326Ala, and Trp330Ala mutations only decreased the potency and efficacy of dobutamine (Fig. 3d). Thr203Ala mutation only decreased the potency and

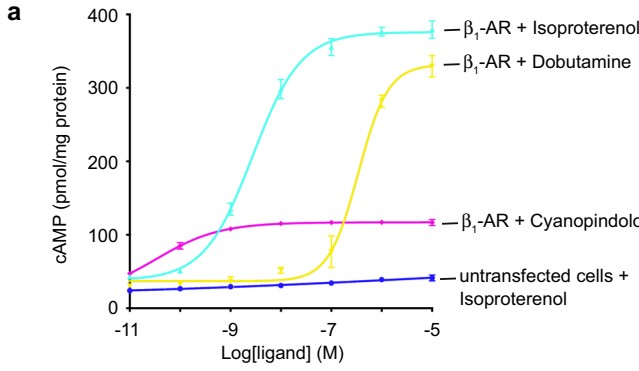

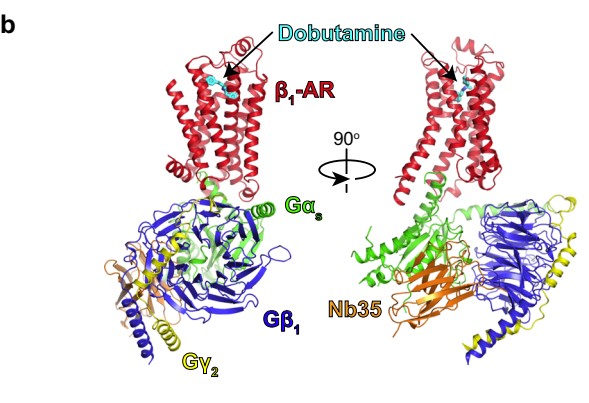

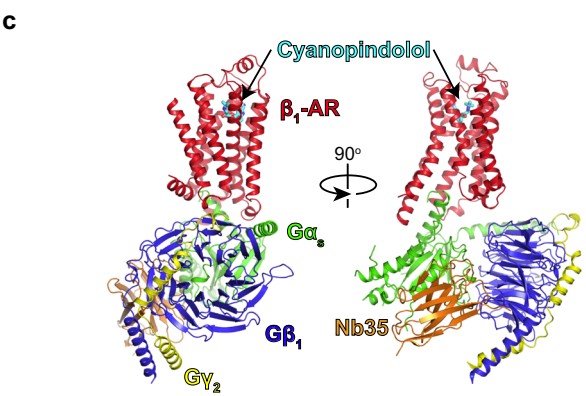

**Fig. 1 Cryo-EM structures of the complexes of Gs and $\beta_1$-AR bound with a partial agonist or a very weak partial agonist. a** Comparison of the cellular cAMP responses induced by isoproterenol, dobutamine, and cyanopindolol. Data are presented as mean ± SD of three experiments. Source data are provided as a Source Data file. **b** Cartoon diagrams of the dobutamine–$\beta_1$-AR–Gs complex are shown. **c** Cartoon diagrams of the cyanopindolol–$\beta_1$-AR–Gs complex are shown. $\beta_1$-AR in red, Ras-like GTPase domain of G$\alpha_s$ in green, G$\beta$ in blue, G$\gamma$ in yellow, and Nb35 nanobody in orange.

efficacy of cyanopindolol (Fig. 3f). These functional data supports the above structural studies identifying these residues involved in specific interactions with the different ligands.

**$\beta_1$-AR conformations within the three $\beta_1$-AR–Gs complexes.** Given our focus on the activation of G-proteins by a GPCR after bound with ligands of different efficacies, we examined whether the $\beta_1$-AR conformations in the three $\beta_1$-AR–Gs complexes are different (Supplementary Fig. 10). At the cytoplasmic side of the receptors, GPCR activation is generally characterized by the displacement of TM5, TM6 and TM7[4] (Supplementary Fig. 10a). Analysis of the three $\beta_1$-AR structures shows that $\beta_1$-ARs in the

three complexes have similar overall conformations with local differences. The conformation in the intracellular half of the TM bundle is notably shifted towards that seen in the active state $\beta_1$-AR structure[20] (Supplementary Fig. 10a), and distinct from those of the inactive-state $\beta_1$-AR structures[26,27] (Supplementary Fig. 10a). In addition to these TM conformational changes, GPCR activations are characterized by the rotameric changes of several conserved motifs[28]. We compared the rotamer positions in the $\beta_1$-ARs in these three complexes and in the inactive state $\beta_1$-AR (Supplementary Fig. 10b–e). Residues Pro219[5.50], Ile129[3.40] and Phe299[6.44] form an interface between TM5, TM3 and TM6 near the base of the ligand binding pocket in $\beta_1$-AR and other class A GPCRs. In the active state structures of $\beta_1$-AR, a chain of conformational rearrangements occur in these residues, in which an inward shift of Phe219[5.50] is coupled with a rotamer switch in Ile129[3.40], a large movement of the Phe299[6.44] side chain, and a corresponding rotation of TM6 on the cytoplasmic side (Supplementary Fig. 10b). All three $\beta_1$-AR structures display similar conformational changes of these residues; no intermediate conformations are observed in the presence of partial agonists (Supplementary Fig. 10b).

Another important aspect of class A GPCR activation is the rearrangement of side chains in highly conserved motifs D(E)/RY (TM3) and NPxxY (TM7), which are referred to as "micro-switches"[28]. The ionic-lock salt bridge is preserved between the side chains of Arg139[3.50] and Asp138[3.49] in the $\beta_1$-AR inactive state, but it is broken in the active state structure (Supplementary Fig. 10c). Additionally, Arg139[3.50] forms a salt bridge with Glu285[6.30] in the inactive state of $\beta_1$-AR[27], but this interaction is broken in the active state of $\beta_1$-AR (Supplementary Fig. 10c). In the active state structure, the Asp138[3.49] side chain forms a hydrogen bond to Tyr149 in ICL2, and the Arg139[3.50] side chain interacts with Tyr377 in the $\alpha$5-helix of G$\alpha_s$ (Supplementary Fig. 10c). The highly conserved NPxxY motif at the cytoplasmic end of TM7 is another key micro-switch of GPCR activation[28]. All three $\beta_1$-AR structures show active state conformations of the NPxxY motif when compared to the inactive $\beta_1$-AR (Supplementary Fig. 10d, e). Therefore, $\beta_1$-ARs in the three different complexes, with Gs-proteins, adopt similar active state conformations, even though they are bound with ligands with different efficacies.

Each of the above three $\beta_1$-AR–Gs complex structures represents the mean conformation of the imaged particles. Three-dimensional variability analysis (3DVA) revealed that, in all three complexes, both $\beta_1$-AR and Gs show conformational flexibility (Supplementary Movies 1-3 for the complex of $\beta_1$-AR–Gs with dobutamine)[29]. Supplementary Movie 1 shows the oscillating movement of $\beta_1$-AR away or towards Gs. Supplementary Movie 2 shows the twisting of $\beta_1$-AR along the membrane axis. Supplementary Movie 3 shows the transverse bending of $\beta_1$-AR, as well as motions of the extracellular parts of $\beta_1$-AR. The N-terminal coiled coil of G$\beta\gamma$ is very dynamic (Supplementary Movies 2 and 3). These types of motions are also seen in the complexes of $\beta_1$-AR–Gs with isoproterenol or cyanopindolol. These analyses reveal the dynamic nature of $\beta_1$-AR and Gs-proteins, as well as their interactions.

GaMD simulations also showed different local conformational flexibilities in the three complexes (Fig. 4, Supplementary Fig. 11). Overall, $\beta_1$-AR underwent small fluctuations in all three complexes except for higher flexibilities in ICL1 and H8[21] (Supplementary Fig. 11a–c). Consistent with their experimental binding affinities, cyanopindolol displayed the lowest fluctuation, while dobutamine with the highest fluctuation (Fig. 4, Supplementary Fig. 12a–c). Within the three complexes, Gs-proteins exhibited higher fluctuations than membrane-embedded $\beta_1$-ARs (Supplementary Fig. 11a–c). The $\alpha$5-helix, the $\alpha$4-$\beta$5 loop, the

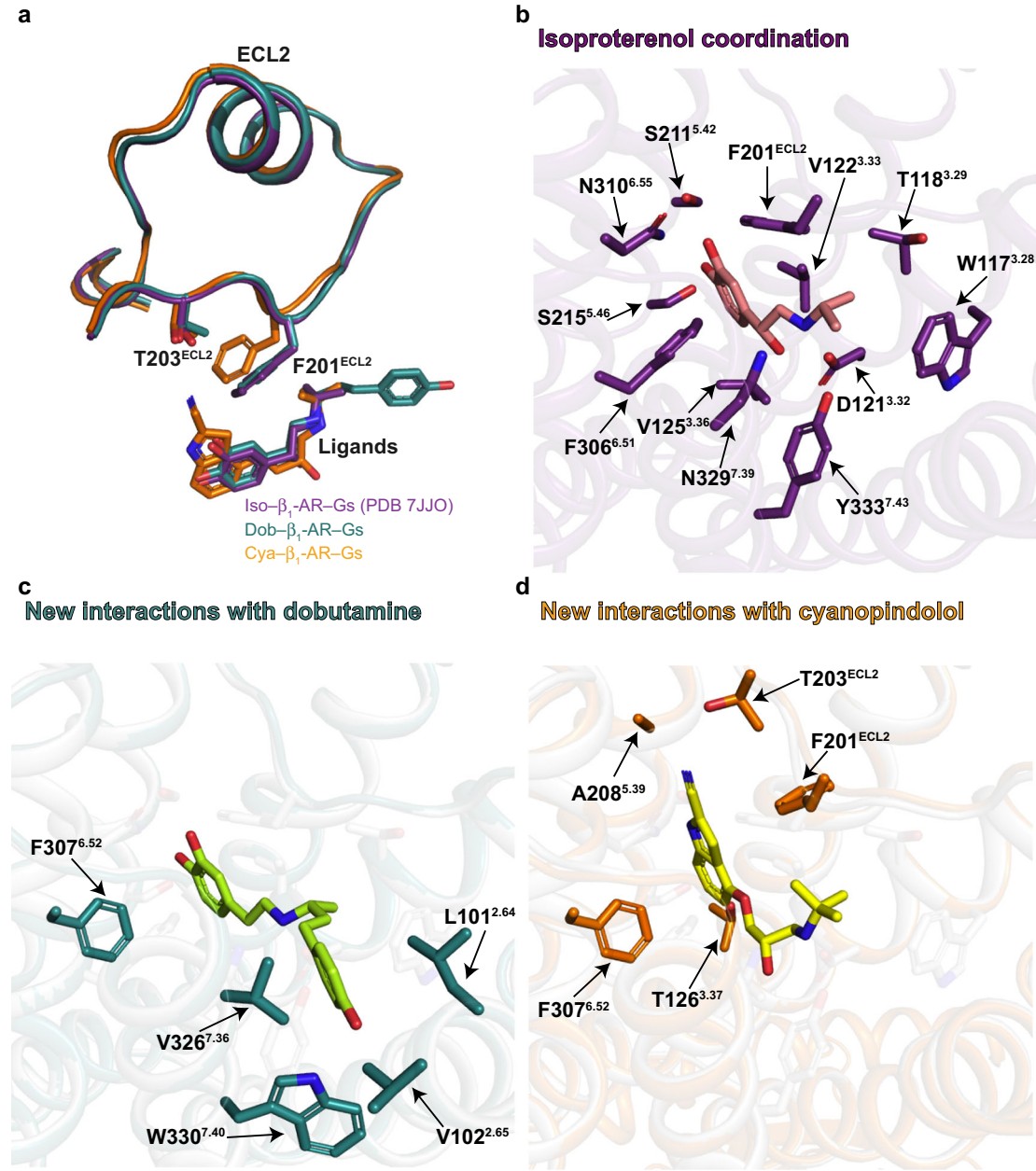

**Fig. 2 Local conformational differences among the three complexes. a** ECL2 segments from the three complexes are shown. While Phe201[ECL2] is involved in the binding to all three ligands, Thr203[ECL2] is only engaged in cyanopindolol interaction. The color codes: Purple: Iso-$\beta_1$-AR–Gs; Green: Dob-$\beta_1$-AR–Gs; Orange: Cya-$\beta_1$-AR–Gs. **b** Diagram of the ligand-binding residues in the isoproterenol-bound $\beta_1$-AR–Gs complex. **c** Additional ligand-binding residues in the dobutamine-bound $\beta_1$-AR–Gs complex, comparing with the isoproterenol-bound $\beta_1$-AR–Gs complex. **d** Additional ligand-binding residues in the cyanopindolol-bound $\beta_1$-AR–Gs complex, comparing with the isoproterenol-bound $\beta_1$-AR–Gs complex.

Switch III region, and the αN-helix of Gα$_s$, as well as the N-termini of Gβγ showed higher structural flexibilities (Supplementary Fig. 11a–c). Compared with the isoproterenol–$\beta_1$-AR–Gs, the dobutamine-$\beta_1$-AR–Gs structure showed different flexibilities in the ligand-binding pocket, TM1, ICL1, TM2, ICL2, TM4, ECL3, TM7 and H8 of $\beta_1$-AR, as well as local regions of Gα$_s$ and Gβγ (Fig. 4b). The cyanopindolol–$\beta_1$-AR–Gs complex also showed different flexibilities in the ligand-binding pocket, TM1, ICL1, ICL2, TM5, TM6, TM7 and H8 of $\beta_1$-AR, as well as local regions of Gα$_s$ and Gβγ (Fig. 4c). Overall, the cyanopindolol–$\beta_1$-AR–Gs complex is relatively less flexible (Fig. 4c). The residues in $\beta_1$-AR that are involved in Gs interactions showed varied flexibilities in the three complexes (Fig. 4d–f). In addition, we simulated $\beta_1$-AR bound by the three agonists after removing Gs from the cryo-EM structures (Supplementary Figs. 11d–f, 12d–f). In the absence of Gs, $\beta_1$-AR displayed higher fluctuations in the ligand-binding pocket, ICL2, the cytoplasmic ends of TM5 and TM6, as well as H8, in all three complexes (Supplementary Fig. 11d–f). Furthermore, isoproterenol and dobutamine underwent higher fluctuations, while the high affinity ligand cyanopindolol remained stable (Supplementary Figures 11 d-f and 12 d-f). With the G-protein, isoproterenol became stabilized, consistent with the allosteric stabilization of agonist binding by G-proteins (Supplementary Fig. 11a–c)[30]. Overall, the GaMD simulations support our cryo-EM structural data showing local conformational differences among the three complexes.

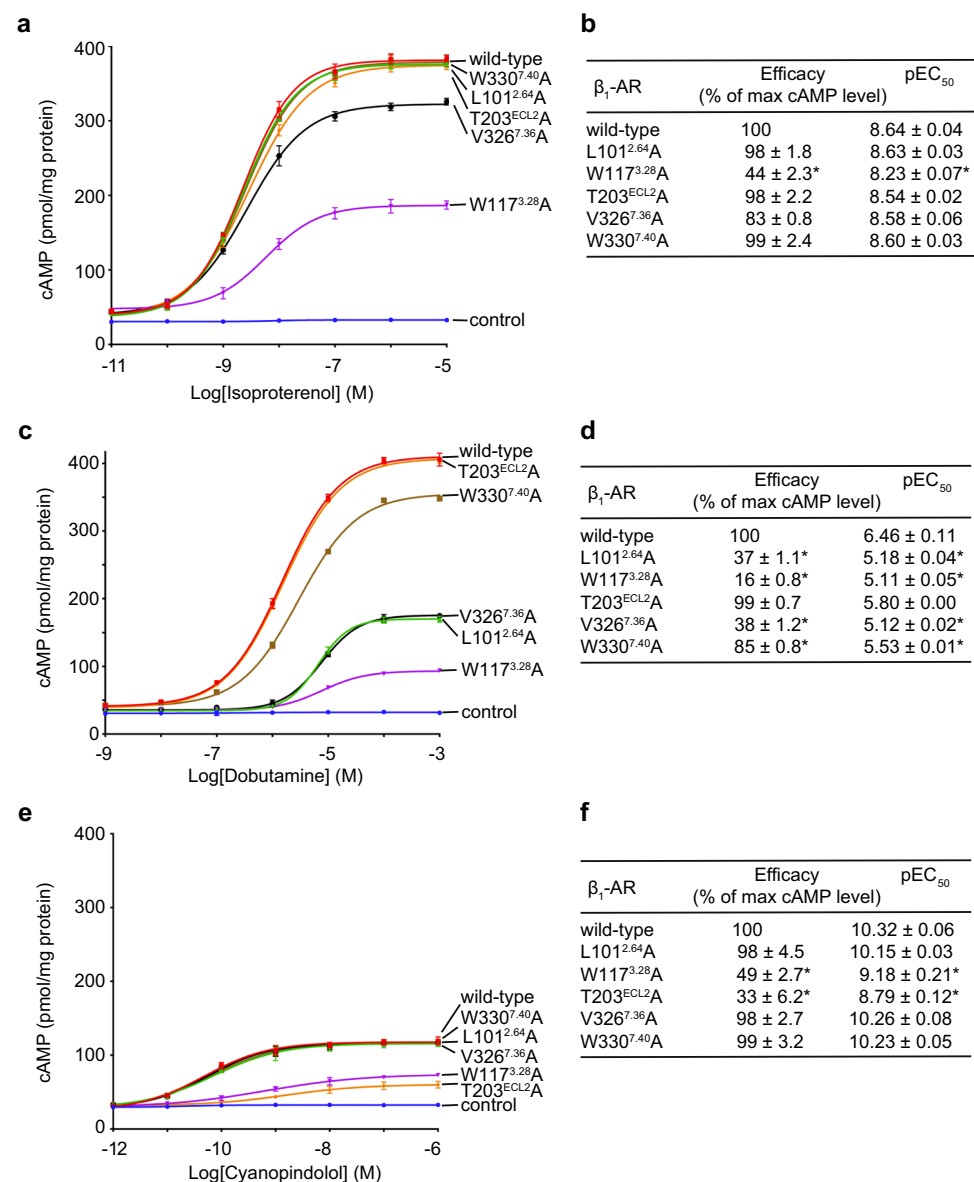

**Fig. 3 Functional studies of receptor residues involved in specific interactions with the different ligands. a, b** Effects of receptor mutations on isoproterenol-induced cellular cAMP responses. **a** Dose–response data from cells expressing different $\beta_1$-AR constructs after stimulation with isoproterenol. **b** Summary of the efficacy (the maximum cAMP level of a mutant receptor / the maximum cAMP level of the wild-type receptor) and the potency ($EC_{50}$ values) based on the cAMP assay data shown in (**a**). **c, d** Effects of receptor mutations on dobutamine-induced cAMP responses. (**c**) Dose–response data from cells expressing different $\beta_1$-ARs after stimulation with dobutamine. (**d**) Summary of the efficacy and $EC_{50}$ values based on the cAMP assay data shown in (**c**). (**e, f**) Effects of mutations on cyanopindolol-induced cAMP responses. (**e**) Dose–response data from cells expressing different $\beta_1$-ARs after stimulation with cyanopindolol. **f** Summary of the efficacy and $EC_{50}$ values based on the cAMP assay data shown in (**e**). Data are shown as mean ± SD of three experiments. The analysis was done using the log(agonist) vs. response function of Prism 8 (GraphPad). Statistical analysis was used to compare individual mutant receptors with the wild-type receptor. *$p < 0.05$ (Student's $t$-test, two-sided). Source data are provided as a Source Data file.

## Cellular studies of the activation of Gs by $\beta_1$-ARs initiated by the three different ligands

We mutated residues on $\beta_1$-AR that participate in its interaction with Gs, and investigated whether these interacting residues contribute similarly or differently to the signaling from $\beta_1$-AR to the cAMP response, after stimulation with the three different ligands (Fig. 5, Supplementary Fig. 9). We selected representative residues from TM5, TM6 and ICL2 since these regions contribute most to the interactions. For isoproterenol, mutations of the interacting residues in $\beta_1$-AR reduced the magnitude of the cAMP response by 37–66% and the $EC_{50}$ by 2 to 6-fold, confirming the importance of these interacting residues for $\beta_1$-AR signaling to Gs[20] (Fig. 5a, b). Similarly,

the $\beta_1$-AR mutants decreased the dobutamine-initiated cAMP response by 17–44% and the $EC_{50}$ by 2 to 7-fold (Fig. 5c, d). Despite cyanopindolol inducing a maximum cAMP response that is ~24% of the cAMP response induced by isoproterenol in the wild-type context, the efficacy and potency were both decreased by the $\beta_1$-AR mutants (Fig. 5e, f). These data indicate that the interacting residues are critical for activation of Gs (and thus the signaling to the downstream cAMP response) by $\beta_1$-AR in response to all three ligands with different efficacies. However, there are different degrees of impairments by some of these mutants (Fig. 5). For example, Gln237[5.68]Ala and Thr291[6.36]Ala had a larger effect on dobutamine-induced cAMP signaling than

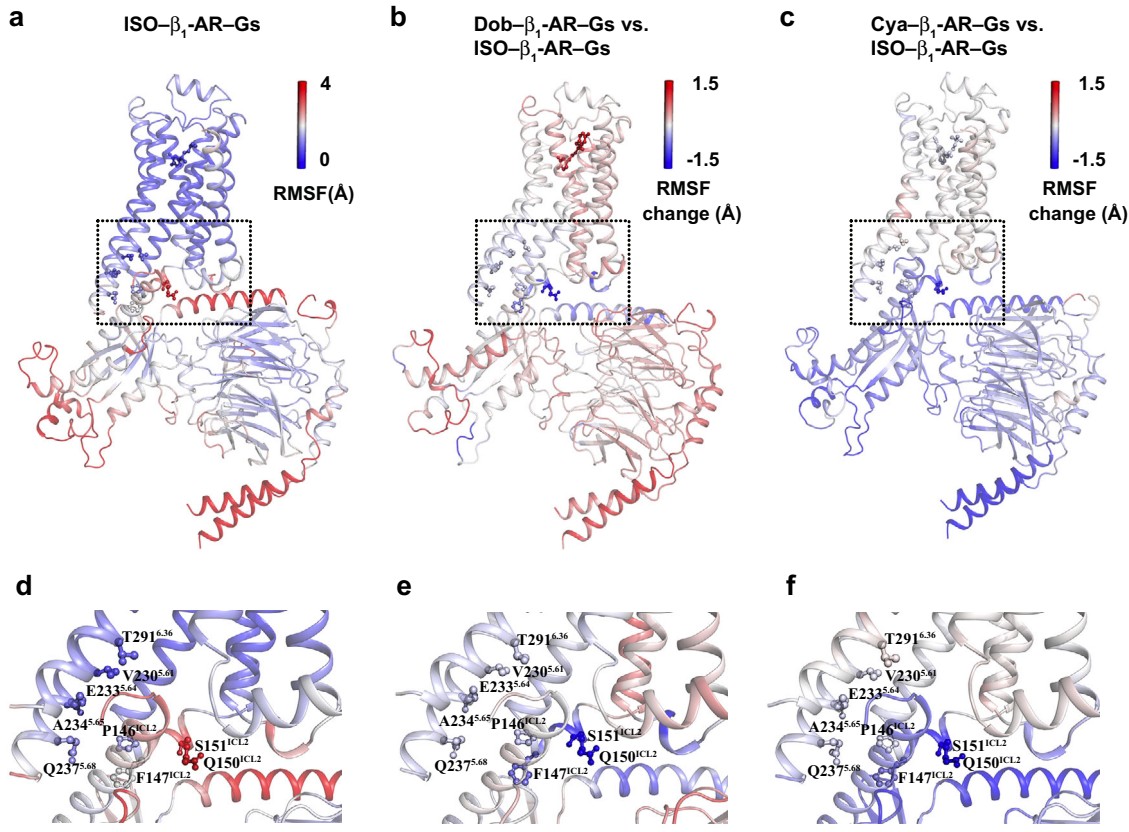

**Fig. 4 Flexibility changes of the agonist-β₁-AR-Gs complexes observed in GaMD simulations. a, d** The root-mean-square fluctuations (RMSFs) of the isoproterenol-β₁-AR-Gs complex. **b, e** Changes in RMSFs of β₁-AR and Gs when the dobutamine-β₁-AR-Gs complex compared with the isoproterenol-β₁-AR-Gs complex. **c, f** Changes in RMSFs of β₁-AR and Gs when the cyanopindolol-β₁-AR-Gs complex compared with the isoproterenol-β₁-AR-Gs complex.

on isoproterenol initiated responses (Fig. 5b, d, f, Supplementary Fig. 9). Phe147^ICL2Ala had a major effect on the cAMP responses stimulated by isoproterenol and dobutamine, than by cyanopindolol (Fig. 5b, d, f, Supplementary Fig. 9). These results indicate that there are residue-specific differences in the signaling mechanisms when bound with ligands with different efficacies.

Furthermore, in addition to the above concentration-dependent cAMP responses, we also investigated the effect of these same mutations on the kinetics of cAMP signaling through β₁-AR (Fig. 6, Supplementary Fig. 13). We measured cAMP responses over time in the presence of near-saturating concentrations (EC₉₀) of isoproterenol, dobutamine, and cyanopindolol (Fig. 6). Isoproterenol induced a quick and robust signal activation phase, and a fast signal termination phase, followed by a high sustaining plateau phase (Fig. 6a). Dobutamine generated a slower signal activation phase and an even slower signal termination phase (Fig. 6b). Cyanopindolol produced a fast activation phase and a fast termination phase without a sustaining phase (Fig. 6c). Mutations of the interacting residues decreased all phases of the signaling responses (Fig. 6, Supplementary Fig. 13). We noticed that there are ligand-specific differences by some of these mutants affecting the rates of activation or termination (Supplementary Fig. 13). For example, Gln237^5.68Ala had a larger effect on the rate of activation by cyanopindolol than by isoproterenol (Supplementary Fig. 13). Pro146^ICL2Ala and Phe147^ICL2Ala had similar effects on the rates of termination by isoproterenol and cyanopindolol, but they had different effects on the rates of termination by dobutamine (Supplementary Fig. 13). We should note that the desensitization phase also depends on the receptor interaction with other proteins (such as G-protein-coupled receptor kinases and arrestins) in cells. These

data affirm that various aspects of the downstream signaling induced by ligands with different efficacies are affected by impairing the β₁-AR and Gs interactions.

**Biochemical studies of the stability of the intermediate state complexes of β₁-AR and Gs when bound with different ligands.** Finally we used biochemical studies to investigate the activation of Gs by β₁-ARs when bound with ligands of different efficacies. GPCRs are enzymes that catalyze the guanine-nucleotide exchange on G-proteins[31]. The nucleotide-free state resolved in the structures with isoproterenol, dobutamine, and cyanopindolol represents an intermediate state in the guanine-nucleotide exchange reaction coordinate[32] (Fig. 7a), and thus we hypothesized that differences in the free energy, and thus stability, of the complex would have profound effects on the activation of G-proteins[33,34]. To investigate whether ligands with different efficacies have different effects on the stability of the intermediate state, we prepared the complex of ligand-bound β₁-AR-Gs (nucleotide-free), and quantified the stability of the intermediate state complex (Fig. 7a). Fluorescently labeled BODIPY-GTPγS binding to the ligand-bound β₁-AR–Gs (nucleotide-free) complex eventually leads to the dissociation of the complex to produce the fluorescently labeled product Gαs(BODIPY-GTPγS), which is detected by an increase in fluorescence (the β₁-AR–Gαs(BODIPY-GTPγS)–Gβγ intermediate is very transient)[31,35] (Fig. 7a–d). The data were then fitted by nonlinear association analyses and half-life values ($t_{1/2} = \ln 2/k$) were calculated based on the rate constant (Fig. 7b–e). The sustained fluorescence levels during the assay time period reflect the known slow dissociation rate of BODIPY-GTPγS from the free Gα subunit[36–38]. The isoproterenol–β₁-AR–Gs complex has the longest half-life and is thus

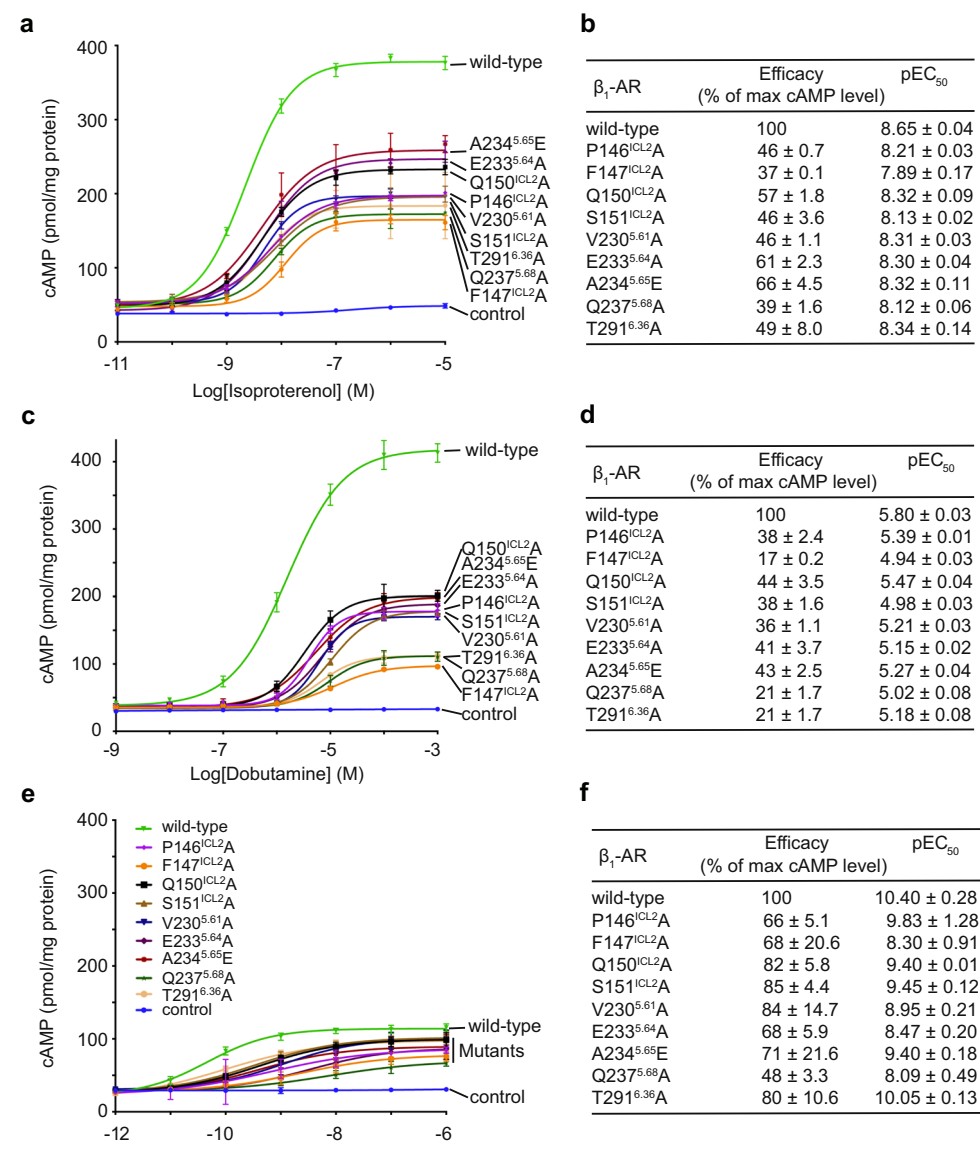

**Fig. 5 Functional studies of the signaling from β₁-ARs initiated by a full agonist, a partial agonist, or a very weak partial agonist. a, b** Effects of receptor mutations on isoproterenol-induced cAMP responses. **a** Dose–response data from cells expressing different β₁-AR constructs after stimulation with isoproterenol. **b** Summary of the efficacy (the maximum cAMP level of a mutant receptor/the maximum cAMP level of the wild-type receptor) and EC₅₀ values based on the cAMP assay data shown in (**a**). **a** and **b** are adapted from[20] and used here for direct comparison. **c, d** Effects of mutations on dobutamine-induced cAMP responses. **c** Dose–response data from cells expressing different β₁-ARs after stimulation with dobutamine. **d** Summary of the efficacy and EC₅₀ values based on the cAMP assay data shown in (**c**). **e, f** Effects of mutations on cyanopindolol-induced cAMP responses. **e** Dose–response data from cells expressing different β₁-ARs after stimulation with cyanopindolol. **f** Summary of the efficacy and EC₅₀ values based on the cAMP assay data shown in (**e**). Data are shown as mean ± SD of three experiments. The analysis was done using the log(agonist) vs. response function of Prism 8 (GraphPad). When comparing with the wild-type receptor, all mutant receptors showed significant difference with *p* values < 0.05 (Student's *t*-test, two-sided). The color keys for the receptor mutants are the same for (**c**) and (**e**), and are displayed in (**e**). Source data are provided as a Source Data file.

the most stable among the three complexes (Fig. 7e). The cyanopindolol–β₁-AR–Gs complex is the least stable, displaying the shortest half-life (Fig. 7e). The dobutamine–β₁-AR–Gs complex displays an intermediate stability (Fig. 7e).

The stability of ligand-bound β₁-ARs in a G-protein compatible conformation (the residence time of β₁-AR in the activate state) is further investigated by GaMD simulations (Fig. 7f–h). As mentioned before, the main conformational change of β₁-AR during its activation is the outward movement of the cytoplasmic end of TM6 (Supplementary Fig. 10). Thus the distance between the cytoplasmic ends of TM3 and TM6 (measured by the distance

between Arg139³·⁵⁰ and Leu289⁶·³⁴) can be used as a measurement of the activation status of β₁-AR[39]. Removal of Gs from the ligand–β₁-AR–Gs complex leads to the deactivation of β₁-AR; this is reflected by the decreased TM3-TM6 distance in GaMD simulations of the cyanopindolol–β₁-AR complex (Fig. 7h, Supplementary Fig. 14i) and the dobutamine–β₁-AR complex (Fig. 7g, Supplementary Fig. 14h). β₁-ARs in these two complexes were more dynamic than in the isoproterenol-bound form (Fig. 7f–h, Supplementary Fig. 14g–i). With isoproterenol, β₁-AR mostly adopted a state with a TM3-TM6 distance of ~12–14 Å (Fig. 7f, Supplementary Fig. 14g). When bound with

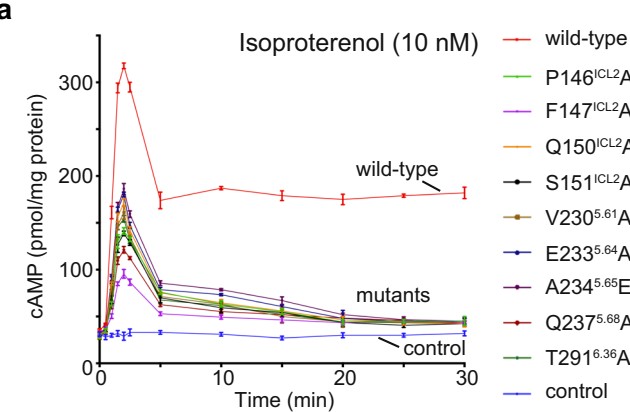

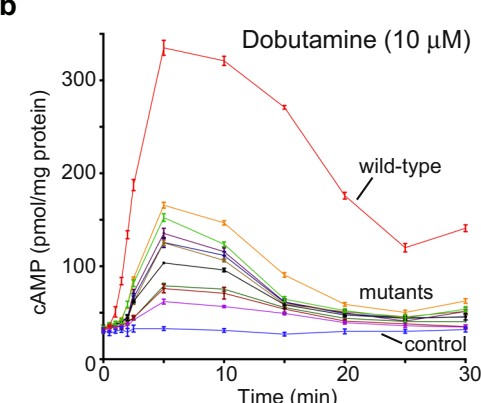

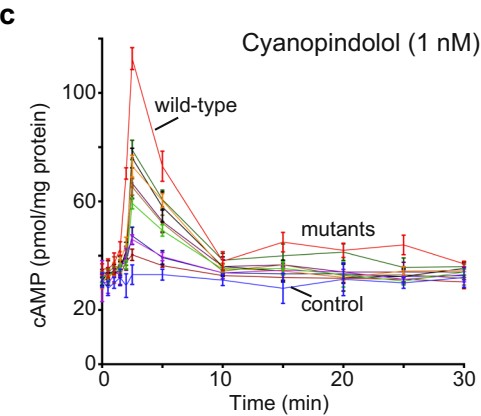

**Fig. 6 Effects of $\beta_1$-AR mutations on the time-dependent cellular cAMP responses initiated by three different ligands.** Time-dependent cAMP responses induced by isoproterenol (**a**), dobutamine (**b**), or cyanopindolol (**c**), are decreased by the $\beta_1$-AR mutations. Data are presented as mean ± SD of three experiments. When comparing with the wild-type receptor, all mutant receptors showed significant difference with $p$ values <0.05 (Student's $t$-test, two-sided). The color keys for the receptor mutants are displayed in (**a**), and are the same for (**a**, **b**, and **c**). Source data are provided as a Source Data file.

dobutamine, $\beta_1$-AR transitioned to the inactive state with a TM3-TM6 distance of ~8.3 Å in one of the three GaMD simulations (Fig. 7g, Supplementary Fig. 14h). Cyanopindolol-bound $\beta_1$-AR transitioned to the inactive state within ~400–700 ns in all three GaMD simulations (Fig. 7h, Supplementary Fig. 14i). These GaMD simulations thus reveal that ligands with higher efficacies are able to maintain $\beta_1$-AR in the active state for longer time. These data are consistent with our above biochemical data.

## Discussion

Our data shows that the structures of the same GPCR–G-protein complexes bound with ligands of different efficacies have overall similar configurations with local conformational differences. From our cryo-EM structures, $\beta_1$-ARs in the three complexes are all in the fully active state. We did not observe a partial activation process in the critical activation microswitch residues in the complexes with a partial agonist or a very weak partial agonist. There are local conformational differences, for example, in the ligand-binding pocket, TM1, ICL1, TM2, ICL2, TM4, ECL3, TM7 and H8 of $\beta_1$-AR. Residue specific differences were confirmed by functional studies using mutant $\beta_1$-ARs in the ligand-binding pockets and in Gs-interaction regions. In the absence of G-proteins, $\beta_1$-ARs bound with ligands with different efficacies were in the inactive state[40]. As shown here, in the presence of G-proteins, $\beta_1$-ARs bound with ligands of different efficacies were in the active state. The conformational changes from the inactive to the active state of $\beta_1$-ARs for all three ligands are similar to other Class A GPCRs (Supplementary Fig. 10). When bound to $\beta_1$-ARs in the presence of Gs or a conformation-specific nanobody, agonists bind tighter with lower RMSDs when compared with the structures without G-proteins[25] (Supplementary Fig. 12). While our manuscript under review, cryo-EM structures of several GPCR–G-protein complexes in the presence of partial agonists were solved, and the overall structures were similar to the full agonist-bound complexes, with some local conformational differences[41–43]. These are consistent with our observations here. Our cellular functional studies show that the residues on $\beta_1$-ARs that interact with Gs are critical for the activation of Gs and the downstream cAMP response since mutations of these residues decreased the efficacy and potency of the cAMP response initiated by isoproterenol, dobutamine and cyanopindolol. We should note that there are some residue-specific differences in their effects on cAMP response initiated by different ligands. These might reflect the local structural differences and the different stabilities of the three complexes.

Our data suggests that the efficacy of the ligand-bound GPCR in catalyzing G-protein activation is also correlated with the stability of the intermediate state of the ligand–GPCR–G-protein complex, which is a complement to the conformation selection model. Our biochemical studies and GaMD simulations show that a full agonist-bound GPCR–G-protein intermediate state complex is more stable than a partial agonist-bound GPCR–G-protein intermediate state complex. Our observation on the stability of the entire ligand–GPCR–G-protein complex is consistent with previous fluorescence spectroscopy experiments showing that a full agonist stabilized the binary complex of $\beta_2$-AR–Gs(nucleotide free) better than a partial agonist[44]. Recently, it has been shown that positive allosteric modulators increase the agonist and receptor (adenosine $A_1$ receptor) efficacy by stabilizing the ligand–GPCR–G protein complex[39]. Future investigations should integrate the thermodynamic and kinetic reaction controls of G-protein activations by GPCRs.

## Methods

**Expression and purification of $\beta_1$-AR, G$\alpha_s$, G$\beta_1$, G$\gamma_2$ and Nb35.** $\beta_1$-AR protein was purified as described previously[20,27]. The turkey $\beta_1$-AR construct $\beta_1$-AR(H12) used in this study was similar to the functional $\beta_1$-AR(H0) construct described previously with some modifications[20,27]. A signal peptide, FLAG tag, PreScission protease cleavage site and T4 lysozyme were fused to the N- terminus with a double-alanine linker, and another PreScission protease cleavage site and His$_6$ tag were added to the C-terminus. $\beta_1$-AR was expressed and purified from Sf9 insect cells grown in ESF 921 protein-free medium (Expression Systems)[20]. Cells were grown to 2–3 million cells per ml before 100 ml of baculoviruses were added for infection. 48 hrs later, cells were harvested by centrifugation, flash frozen in liquid nitrogen and stored at −80 °C until use. For membrane preparation, cell pellets were lysed by sonication in a buffer containing 20 mM Tris, pH 8, 1 mM EDTA and protease inhibitor cocktail (Sigma) and washed once more using the same

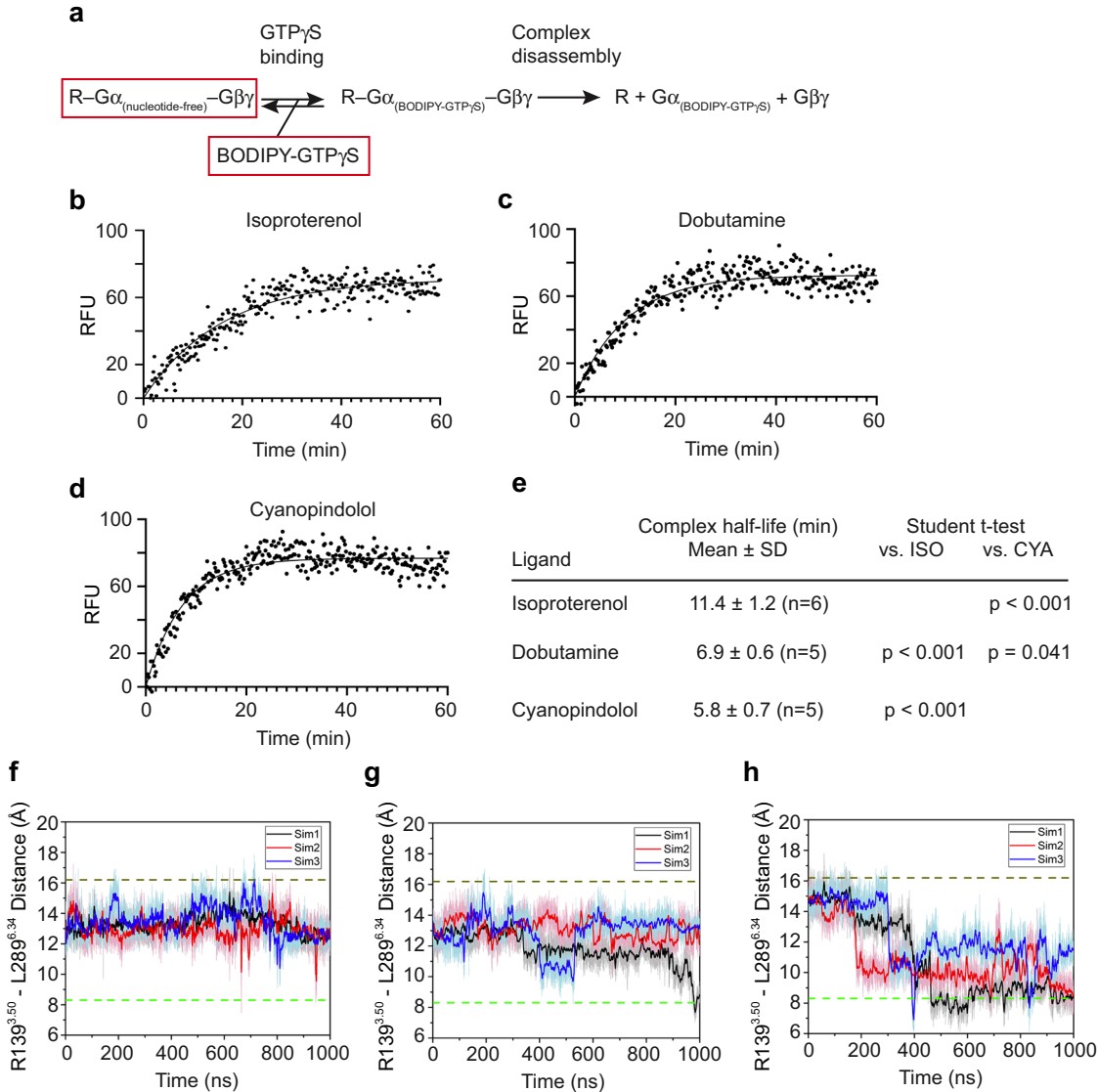

**Fig. 7 Biochemical studies and GaMD simulations of the stability of the complexes of β₁-AR and Gs when bound with a full agonist, a partial agonist, or a very weak partial agonist. a** A schematic diagram represents the chemical process from BODIPY-GTPγS binding to the complex of ligand–receptor–Gα(nucleotide-free)–Gβγ, leading to the formation of the transient R-G(BODIPY-GTPγS) bound complex and the subsequent complex disassembly. **b–e** BODIPY-GTPγS binding to the transition state complex in the presence of isoproterenol (**b**), dobutamine (**c**), or cyanopindolol (**d**). Units reported as relative fluorescent units (RFU). One representative experiment from five or six independent experiments with similar results is shown for each case. **e** Summary of the half-life values. Data are shown as mean ± SD of five or six independent experiments. Two-sided *P* values are from Student's *t*-test. **f–h** Ligand-dependent structural dynamics of β₁-AR in the absence of Gs. The distance between the cytoplasmic ends of TM3 and TM6 (measured as the distance between the Cα atoms of Arg139³·⁵⁰ and Leu289⁶·³⁴) is calculated over the indicated time. Three GaMD simulations (black, red, blue) were performed. Lines depict the running average over 2 ns. The top dash lines indicate the distance observed in the cryo-EM structures, and the bottom dash lines indicate the distance observed in the inactive β₁-AR (PDB 4GPO).

buffer. Purified membranes were resuspended in 20 mM Tris, pH 8, 0.2 mM EDTA, and protease inhibitor cocktail and flash frozen in liquid nitrogen and stored at −80 °C. For protein purification, membrane preparations were first thawed in 20 mM Tris, pH 8, 350 mM NaCl, and protease inhibitor cocktail. 1 mM isoproterenol (Sigma) was then added and the mixture was stirred for 1 h at 4 °C and the membranes were then solubilized in 20 mM Tris, pH 8, 350 mM NaCl, 1% n-Dodecyl-β-D-Maltopyranoside (DDM, Anatrace), 1 mM isoproterenol and protease inhibitor cocktail for 1 hr at 4 °C. The DDM concentration was then reduced to 0.5% by adding equal volume of 20 mM Tris, pH 8, 350 mM NaCl, and 1 mM isoproterenol and the mixture was stirred for another 1 hr at 4 °C. The preparation was clarified by ultracentrifugation at 142,000 g for 30 min at 8 °C. The supernatant was then incubated with Ni-NTA resin (Qiagen) with stirring at 4 °C with 8 mM imidazole. After 4 hrs, the resin was collected by centrifugation and washed three times with 20 mM Tris, pH 8, 500 mM NaCl, 0.05% DDM, 1 mM isoproterenol, and 20 mM imidazole and one time with 20 mM Tris, pH 8, 150 mM NaCl, 0.05% DDM, 1 mM isoproterenol, and 20 mM imidazole. β₁-AR was then eluted from the resin with 20 mM Tris, pH 8, 150 mM NaCl, 0.05% DDM, 1 mM

isoproterenol, and 200 mM imidazole. The elution was concentrated and further purified by size-exclusion chromatography using a Superdex 200 Increase 10/300 column (GE Healthcare) pre-equilibrated with 20 mM Tris, pH 8, 150 mM NaCl, 0.02% Lauryl Maltose Neopentyl Glycol (LMNG, Anatrace), 1 mM isoproterenol. Dobutamine- and cyanopindolol-bound β₁-AR proteins were purified using the same protocol with 200 μM dobutamine and 50 μM cyanopindolol present during purification. Purified β₁-AR was concentrated to 4 mg/ml and either used immediately for complex assembly or flash frozen in liquid nitrogen and stored at −80 °C.

The recombinant wild-type bovine Gαs was purified from *E. coli* strain BL21(DE3)[20,45]. This Gαs construct had an N-terminal GST tag that was removable through a PreScission protease cleavage site. Cells were grown in 2 × YT medium at 37 °C until OD₆₀₀ reached 0.6. Protein expression was then induced by 75 μM IPTG and continued for 16 h at 16 °C. Cells were harvested by centrifugation, flash frozen in liquid nitrogen and stored at −80 °C. For protein purification, cell pellets were thawed in a lysis buffer containing 20 mM HEPES, pH 7, 150 mM NaCl, 10% glycerol, 5 mM β-mercaptoethanol, 2 mM MgCl₂, 1 mM EDTA, 10 μM GDP,

0.1 mg/ml lysozyme, 0.2 mM PMSF and protease inhibitor cocktail, and further lysed by sonication. Cell debris was removed by centrifugation at 20,000 g for 40 min 4 °C. Supernatant was then collected and incubated with Glutathione resin (Pierce) with stirring for 1 hr at 4 °C. Resin was then washed four times with 20 mM HEPES, pH 7, 150 mM NaCl, 10% glycerol, 5 mM β-mercaptoethanol, 2 mM MgCl₂, 1 mM EDTA, and 10 μM GDP. To remove the GST tag, PreScission protease was added to the beads at 1:10 (w:w) protease: GST-Gαs ratio and the mixture was rocked overnight at 4 °C with 2 mM DTT. Untagged Gαs was concentrated and further purified by size-exclusion chromatography using a Superdex 200 Increase 10/300 column pre-equilibrated with 20 mM HEPES, pH 7, 150 mM NaCl, 10% glycerol, 5 mM β-mercaptoethanol, 1 mM MgCl₂, 1 mM EDTA, 20 μM GDP. Purified Gαs was concentrated to 6 mg/ml, flash frozen in liquid nitrogen and stored at −80 °C.

Recombinant bovine Gβ₁ and bovine His₆-tagged soluble Gγ₂(C68S) were co-expressed and purified from Sf9 insect cells[20]. 25 ml of each baculovirus were co-infected into Sf9 cells when the insect cell culture reached a cell density at 3 million cells per ml. 48 h post infection, cells were harvested by centrifugation, flash frozen in liquid nitrogen and stored at −80 °C. Cell pellets were thawed in 25 mM HEPES pH 7, 150 mM NaCl, 2 mM β-mercaptoethanol, and protease inhibitor cocktail. Cells were lysed by sonication and cell debris were removed by centrifugation at 142,000 g for 30 min. Supernatant was collected and incubated with Ni-NTA resin with stirring for 1.5 h at 4 °C. Resin was then washed three times with 25 mM HEPES pH 7, 150 mM NaCl, 2 mM β-mercaptoethanol, and 25 mM imidazole, and Gβ₁γ₂ was eluted as a complex with 25 mM HEPES pH 7, 150 mM NaCl, 2 mM β-mercaptoethanol, and 250 mM imidazole. Eluted protein was concentrated and further purified using a Superdex 200 Increase 10/300 column pre-equilibrated with 25 mM HEPES pH 7, 150 mM NaCl, and 2 mM β-mercaptoethanol. Purified Gβ₁γ₂ protein was concentrated to 8 mg/ml, flash frozen in liquid nitrogen and stored at −80 °C.

Nb35-His₆ was expressed in the periplasm of E. coli strain BL21(DE3)[20]. Cells were grown in LB medium at 37 °C until OD₆₀₀ reached 0.6. Protein expression was then induced by 75 μM IPTG and Nb35 was further expressed for 18 h at 16 °C. Cells were then harvested, flash frozen in liquid nitrogen and stored at −80 °C. For protein purification, cells were lysed by sonication in a lysis buffer containing 20 mM HEPES pH 7, 100 mM NaCl, 5 mM MgCl₂, 0.1 mM lysozyme, and protease inhibitor cocktail. After removal of the cell debris by centrifugation at 20,000 g for 30 min, supernatant was collected and incubated with Ni-NTA resin with stirring for 1.5 hrs at 4 °C. Resin was then washed three times with 20 mM HEPES pH 7, 100 mM NaCl, and 25 mM imidazole. Nb35 was eluted with 20 mM HEPES pH 7, 100 mM NaCl, and 250 mM imidazole. Eluted Nb35 protein was dialyzed against 1 L of 20 mM HEPES pH 7, 100 mM NaCl overnight at 4 °C. Dialyzed protein was concentrated to 3 mg/ml, flash frozen in liquid nitrogen and stored in −80 °C.

**Protein complex assembly and purification**. To assemble the β₁-AR-Gs-Nb35 complex with different ligands bound, Gαs, Gβ₁γ₂ and Nb35 were mixed at 1:1:1.5 molar ratios in the presence of 2 mM MgCl₂. The mixture was incubated for 30 min at room temperature and then mixed with β₁-AR at 1.2:1 ratio in the presence of isoproterenol, dobutamine or cyanopindolol. The mixture was diluted with buffer containing 10 mM HEPES pH 7, 100 mM NaCl, 0.1 mM TCEP, 0.02% LMNG, and 2 mM MgCl₂ to bring the volume to 500 μl. The final concentration of three different ligands in the mixture was 1 mM, 200 μM and 50 μM of isoproterenol, dobutamine and cyanopindolol, respectively. This mixture was incubated for another 30 min at room temperature before 0.4 U Apyrase (Sigma) was added. After additional 30 min room temperature incubation with Apyrase, the mixture was centrifuged at 16,000 g for 10 min to remove any precipitants. The supernatant was then loaded onto a Superdex 200 Increase 10/300 column pre-equilibrated with 10 mM HEPES pH 7, 100 mM NaCl, 0.1 mM TCEP, 0.02% LMNG and 40 uM corresponding ligands. The elution fractions from a single peak containing pure β₁-AR-Gs-Nb35 complex was concentrated to ~1.8 mg/ml and used directly for making cryo-EM grids.

**Cryo-EM data collection**. Four microlitre of protein complex was applied to a glow-discharged 400 mesh gold Quantifoil R1.2/1.3 holey carbon grids (Quantifoil Micro Tools), and subsequently vitrified with Vitrobot Mark IV (Thermo Fisher Scientific/FEI). Images were collected at liquid nitrogen temperature on a Titan Krios electron microscope (Thermo Fisher Scientific/FEI) operated at 300 kV accelerating voltage, at a nominal magnification of ×22,500 using a Gatan K3 direct electron detector (Gatan, Inc.) with SerialEM3.7. For cyanopindolol, a total of 10,000 micrographs were collected between −1.0 and −2.3 μm defocus. For dobutamine, a total of 9305 micrographs were collected between −1.0 and −2.3 μm. The improved DQE of the K3 enabled data acquisition at lower accumulated doses, with a final dose of 28 e⁻/Å². The dose rate of 20 e⁻/pix/s was fractionated over 1.5 s into 60 frames.

**Image processing, 3D reconstructions, modeling and refinement**. Full-frame motion correction was performed in Relion 3.1 using MotionCor2[46]. CTF estimation was performed in Relion 3.1 using CTFFind4[47]. Relion 3.1[48] Laplacian-of-Gaussian picking with minimum and maximum dimensions of 76 Å and 119 Å was used to heavily over-pick at a rate of approximately 2300 particles per micrograph.

The resulting particle stacks of 18 million (cyanopindolol) and 17 million (dobutamine) particles was Fourier-cropped and processed through multiple rounds of heterogeneous classification in CryoSparc v2.14.2[49], steadily decreasing the cropping factor as junk was removed and resolution improved (Supplementary Figs. 1 and 2). 2D classification confirmed that the majority of particles were false positives, receptor alone or G-proteins alone. The final stacks of intact complexes were 2.9 million (cyanopindolol) and 2.6 million (dobutamine) particles. Further classification converged on final high-resolution stacks of 657,613 (cyanopindolol) and 440,739 (dobutamine) particles that were then subjected to Local CTF Refinement procedures in CryoSparc v2.14.2 followed by Bayesian Polishing in Relion 3.1, and finally Global CTF Refinement in CryoSparc v2.14.2 to improve higher order aberrations (Supplementary Figs. 1 and 2). Final high-resolution reconstructions were subjected to Local Refinement with Non-Uniform Refinement in CryoSparc v2.14.2 for β₁-AR and G-proteins independently. The Local Refinement maps showed significantly improved features over the consensus maps, both with resolutions better than 2.5 Å (cyanopindolol) and 2.7 Å (dobutamine) Supplementary Figs. 1 and 2). All maps underwent the density modification (Resolve CryoEM) procedure in Phenix dev-3765, further improving the resolution[50] (Supplementary Figs. 1 and 2). The resulting maps were super-sampled in Coot v0.9-pre[51] to 0.71 Å per pixel with a 384-voxel box to bring out features at high resolution. The initial models of β₁-AR, Gαs, Gβ₁γ₂, and Nb35 were derived from the cryo-EM structure of the complex of isoproterenol–β₁-AR–Gs (PDB ID: 7JJO). Concurrently with the data processing, the models were built in Coot v0.9-pre and Real-Space Refined in Phenix dev-3765[52] as resolutions improved, enabling a final composite map to be derived from the model and the two super-sampled local refinement maps using the Combine Focused Maps feature in Phenix dev-3765. Final rounds of Phenix dev-3765 Real-Space Refinement against the final composite map yielded the final published models for cyanopindolol and dobutamine.

**Gaussian accelerated molecular dynamics (GaMD)**. GaMD is an enhanced sampling method that works by adding a harmonic boost potential to reduce the system energy barriers[21,22]. When the system potential $V\left(\vec{r}\right)$ is lower than a reference energy E, the modified potential $V^*\left(\vec{r}\right)$ of the system is calculated as:

$$V^*\left(\vec{r}\right) = V\left(\vec{r}\right) + \triangle V\left(\vec{r}\right)$$

$$\triangle V\left(\vec{r}\right) = \begin{cases} \frac{1}{2}k\left(E - V\left(\vec{r}\right)\right)^2, & V\left(\vec{r}\right) < E \\ 0, & V\left(\vec{r}\right) \ge E, \end{cases} \quad (1)$$

where $k$ is the harmonic force constant. The two adjustable parameters E and k are automatically determined on three enhanced sampling principles. First, for any two arbitrary potential values $v_1\left(\vec{r}\right)$ and $v_2\left(\vec{r}\right)$ found on the original energy surface, if $V_1\left(\vec{r}\right) < V_2\left(\vec{r}\right)$, $\triangle V$ should be a monotonic function that does not change the relative order of the biased potential values; i.e., $V_1^*\left(\vec{r}\right) < V_2^*\left(\vec{r}\right)$. Second, if $V_1\left(\vec{r}\right) < V_2\left(\vec{r}\right)$, the potential difference observed on the smoothened energy surface should be smaller than that of the original; i.e., $V_2^*\left(\vec{r}\right) - V_1^*\left(\vec{r}\right) < V_2\left(\vec{r}\right) - V_1\left(\vec{r}\right)$. By combining the first two criteria and plugging in the formula of $V^*\left(\vec{r}\right)$ and $\triangle V$, we obtain

$$V_{\max} \le E \le V_{min} + \frac{1}{k}, \quad (2)$$

Where $V_{\min}$ and $V_{\max}$ are the system minimum and maximum potential energies. To ensure that Eq. 2 is valid, k has to satisfy: $k \le 1/\left(V_{\max} - V_{\min}\right)$. Let us define: $k = k_0 \cdot 1/\left(V_{\max} - V_{\min}\right)$, then $0 < k_0 \le 1$. Third, the standard deviation (SD) of $\triangle V$ needs to be small enough (i.e. narrow distribution) to ensure accurate reweighting using cumulant expansion to the second order:

$\sigma_{\triangle V} = k\left(E - V_{avg}\right)\sigma_V \le \sigma_0$, where $V_{avg}$ and $\sigma_V$ are the average and SD of $\triangle V$ with $\sigma_0$ as a user-specified upper limit (e.g., $10k_BT$) for accurate reweighting. When E is set to the lower bound $E = V_{\max}$ according to Eq. 2, $k_0$ can be calculated as

$$k_0 = \min\left(1.0, k_0'\right) = \min\left(1.0, \frac{\sigma_0}{\sigma_V} \cdot \frac{V_{\max} - V_{\min}}{V_{\max} - V_{avg}}\right), \quad (3)$$

Alternatively, when the threshold energy E is set to its upper bound $E = V_{\min} + 1/k$, $k_0$ is set to:

$$k_0 = k_0'' \equiv \left(1 - \frac{\sigma_0}{\sigma_V}\right) \cdot \frac{V_{\max} - V_{\min}}{V_{avg} - V_{\min}}, \quad (4)$$

If $k_0''$ is calculated between 0 and 1. Otherwise, $k_0$ is calculated using Eq. 3.

**System setup**. The isoproterenol–β₁-AR–Gs, dobutamine–β₁-AR–Gs and cyano-pindolol–β₁-AR–Gs cryo-EM structures were used for setting up simulation systems. The initial structures of isoproterenol–β₁-AR, dobutamine–β₁-AR and

cyanopindolol–$\beta_1$-AR were obtained by removing $G_s$ from the isoproterenol–$\beta_1$-AR–$G_s$, dobutamine–$\beta_1$-AR–$G_s$ and cyanopindolol–$\beta_1$-AR–$G_s$ cryo-EM structures. According to previous findings, ICL 3 is highly flexible and removal of ICL3 does not appear to affect GPCR function[53,54]. ICL3 missing in the cryo-EM structures was thus omitted in the GaMD simulations. Similarly[55], the α-helical domain of $G_s$ missing in the cryo-EM structures was not included in the simulation models. This was based on earlier simulation of the $\beta_2$-AR–$G_s$ complex, which showed that the α-helical domain fluctuated substantially[53]. Other missing residues in $G_s$ were modelled using SWISS Modeller[56]. All chain termini were capped with neutral groups (acetyl and methylamide). All the disulphide bonds in the complexes that were resolved in the cryo-EM structures were maintained in the simulations. Using the *psfgen* plugin in VMD[57], missing atoms in protein residues were added and all protein residues were set to the standard CHARMM protonation states at neutral pH. For each of the complex systems, the receptor was inserted into a palmitoyl-oleoyl-phosphatidyl-choline (POPC) bilayer with all overlapping lipid molecules removed using the membrane plugin in VMD[57]. The system charges were then neutralized at 0.15 M NaCl using the solvate plugin in VMD[57]. The simulation systems were summarized in Supplementary Table 3.

**Simulation protocol**. The CHARMM36m parameter set[58–60] was used for the proteins and lipids. Force field parameters of the agonists (isoproterenol, dobutamine and cyanopindolol) were obtained from the ParamChem web server[61]. Force field parameters with high penalty were optimized used with FFParm[62]. GaMD simulations of these systems followed a similar protocol as in previous studies of GPCRs[55,63,64]. For each of the complex systems, initial energy minimization, thermalization, and 20 ns cMD equilibration were performed using NAMD2.12[65]. A cutoff distance of 12 Å was used for the van der Waals and short-range electrostatic interactions and the long-range electrostatic interactions were computed with the particle-mesh Ewald summation method[66]. A 2-fs integration time step was used for all MD simulations and a multiple-time-stepping algorithm was used with bonded and short-range non-bonded interactions computed every time step and long-range electrostatic interactions every two timesteps. The SHAKE algorithm[67] was applied to all hydrogen-containing bonds. The NAMD simulation started with equilibration of the lipid tails. With all other atoms fixed, the lipid tails were energy minimized for 1,000 steps using the conjugate gradient algorithm and melted with a constant number, volume, and temperature (NVT) run for 0.5 ns at 310 K. The four systems were further equilibrated using a constant number, pressure, and temperature (NPT) run at 1 atm and 310 K for 10 ns with 5 kcal/(mol· Å²) harmonic position restraints applied to the protein and ligand atoms. Final equilibration of each system was performed using an NPT run at 1 atm pressure and 310 K for 0.5 ns with all atoms unrestrained. After energy minimization and system equilibration, conventional MD simulations were performed on each system for 20 ns at 1 atm pressure and 310 K with a constant ratio constraint applied on the lipid bilayer in the X-Y plane.

With the NAMD output structures, the system topology and CHARMM36m force field files, the *ParmEd* tool in the AMBER package[68] was used to convert the simulation files into the AMBER format. The GaMD module implemented in the GPU version of AMBER20[21,68] was then applied to perform the simulations. GaMD simulations of the isoproterenol–$\beta_1$-AR–$G_s$, dobutamine–$\beta_1$-AR–$G_s$ and cyanopindolol–$\beta_1$-AR–$G_s$ included an 8.5-ns short cMD simulation used to collect the potential statistics for calculating GaMD acceleration parameters, a 68-ns equilibration after adding the boost potential, and finally three independent 500-ns GaMD production simulations with randomized initial atomic velocities. The average and SD of the system potential energies were calculated every 850,000 steps (1.7 ns). GaMD simulations of isoproterenol–$\beta_1$-AR, dobutamine–$\beta_1$-AR and cyanopindolol–$\beta_1$-AR with smaller system sizes included a 2.8-ns short cMD simulation used to collect the potential statistics for calculating GaMD acceleration parameters, a 50.4-ns equilibration after adding the boost potential, and finally three independent 1000-ns GaMD production simulations with randomized initial atomic velocities. The average and SD of the system potential energies were calculated every 280,000 steps (0.56 ns). All GaMD simulations were run at the "dual-boost" level by setting the reference energy to the lower bound. One boost potential was applied to the dihedral energetic term and the other to the total potential energetic term. The upper limit of the boost potential SD, $\sigma_0$ was set to 6.0 kcal/mol for both the dihedral and the total potential energetic terms. Similar temperature and pressure parameters were used as in the NAMD simulations.

**Simulation analysis**. CPPTRAJ[69] and VMD[57] were used to analyze the GaMD simulations. The root-mean square deviations (RMSDs) of the agonists (isoproterenol, dobutamine and cyanopindolol) relative to the cryo-EM structures and the distance between the receptor TM3 and TM6 intracellular ends (measured by the distance between the Cα atoms of receptor residues Arg139$^{3.50}$ and Leu289$^{6.34}$) were selected as reaction coordinates. Time courses of these reaction coordinates obtained from the GaMD simulation were plotted in Fig. 7f–h, Supplementary Figs. 12, 14a–c. Root-mean-square fluctuations (RMSFs) were calculated for the protein residues and agonists, averaged over three independent GaMD simulations and color-coded for schematic representation of each complex system (Fig. 4, Supplementary Fig. 11). The PyReweighting[70] toolkit was applied to reweight GaMD simulations to recover the original free energy profiles of the simulation systems. 2D free energy profiles were computed using the combined trajectories from all the three independent GaMD simulations for each system with agonist RMSD and TM3-TM6 distance as reaction coordinates (Supplementary Figs. 14d–i). A bin size of 1.0 Å was used for agonist RMSD and TM3-TM6 distance. The cutoff was set to 500 frames for 2D free energy calculations.

**cAMP assay**. CHO cells (transfected with a control empty vector or wild-type or mutant turkey $\beta_1$-AR) were plated onto six-well plates, and were pre-incubated with culture medium buffered with 0.5 mM IBMX for 30 min at 37 °C[20]. After washing twice with HEM buffer (20 mM HEPES, pH 7.4, 135 mM NaCl, 4.7 mM KCl, 1.2 mM MgSO₄, 2.5 mM NaHCO₃, 0.1 mM Ro-20-1724, 0.5 U/ml adenosine deaminase, and 1 mM IBMX), cells were treated with different concentrations of ligands in HEM buffer for 5 min for the dose–response studies. For the time course studies, cells were stimulated with ligands (10 nM for isoproterenol, 10 μM for dobutamine, and 1 nM for cyanopindolol) for 0, 0.5, 1, 1.5, 2, 2.5, 5, 10, 15, 20, 25, 30 min at 37 °C. After culture medium removal, cells were treated with 0.1 M HCl for 10 minutes at room temperature. After centrifugation, the supernatant was used for cAMP quantification using the Direct Cyclic AMP Enzyme Immunoassay kit (Enzo Life Sciences). Membrane receptor ($\beta_1$-AR) expressions in these transiently transfected cells were measured by Western blots using a monoclonal anti-$\beta_1$-AR antibody and were found to be at similar levels[20]. The cAMP assays were repeated three times, and the data are represented as mean ± SD of the three independent experiments. The analysis was done using the log(agonist) vs. response function of Prism 8 (GraphPad)[20].

**BODIPY-GTPγS binding assays**. Both BODIPY-GTPγS binding assays were performed in clear plastic 96-well plates and measured using a SpectraMAX Gemini EM microplate reader (Molecular Devices) with excitation at 485 nm and emission read behind a 530 nm longpass filter. In 100 μl binding buffer (10 mM HEPES, pH 7, 100 mM NaCl, 0.1 mM TECP, 0.02% LMNG, 1 mM EDTA, and 2 mM MgCl₂), 200 nM ligand-bound nucleotide-free $\beta_1$-AR–Gs complex, and the ligand at a concentration of ~EC₉₀ (10 nM isoproterenol, 10 μM dobutamine or 1 nM cyanopindolol) were added. The reaction was initiated by adding 10 μM BODIPY™ FL GTPγS (Invitrogen). Relative fluorescence units (RFU) change was measured every 12 s for a total of 60 min at 25 °C. The BODIPY-GTPγS binding data were fitted to one phase exponential association curves using GraphPad Prism 8.

**Quantification and statistical analysis**. In Figs. 3, 5 and 6, the cAMP assays were repeated three times, and the data are represented as mean ± SD of the three independent experiments. The analysis was done using the log(agonist) vs. response function of Prism 8 (GraphPad) as indicated in the figure legends. Cryo-EM data collection and refinement statistics are listed in Supplementary Tables 1 and 2.

**Reporting summary**. Further information on research design is available in the Nature Research Reporting Summary linked to this article.

## Data availability

The cryo-EM density maps and corresponding coordinates have been deposited in the Electron Microscopy Data Bank (EMDB) and the PDB, respectively, under the accession codes: EMD-27328 (dobutamine–$\beta_1$-AR–Gs), EMD-27329 (cyanopindolol–$\beta_1$-AR–Gs), and PDB 8DCR (dobutamine–$\beta_1$-AR–Gs), 8DCS (cyanopindolol–$\beta_1$-AR–Gs). Source data are provided with this paper.

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

## Acknowledgements

We thank members of our research groups for helpful discussion and comments on the manuscript. This work was supported by NIH grants GM138676 (X.Y.H.), DA042298 (W.L.), GM124152 (W.L.), and GM132572 (Y.M.), the Josie Robertson Investigators Program (R.K.H.), and the Searle Scholars Program (R.K.H.). This work used super-computing resources with XSEDE allocation award TG-MCB180049 and NERSC project M2874.

## Author contributions

M.S. expressed and purified $\beta_1$-AR, $G\alpha_s$, $G\beta_1\gamma_2$, Nb35 and the protein complexes, made cryo-EM grids, performed cryo-EM screening, data collection, model building, and BODIPY-GTP$\gamma$S binding studies. N.P. made cryo-EM grids, performed cryo-EM screening, data collection, image processing, EM density map determination, and model building under the supervision of R.K.H. L.Z. performed cAMP assays under the supervision of W.L. J.W. and H.N.D. performed GaMD simulations under the supervision of Y.M. X.Y.H. supervised the project, interpreted data and wrote the manuscript. All authors contributed towards the final version of the manuscript.

## Competing interests

The authors declare no competing interests.
