## [Peer Review File · Nature Communications]

Structures of β 1-Adrenergic Receptor in Complex with Gs and Ligands of Different EfficaciesEditorial Note: This manuscript has been previously reviewed at another journal that is not operating a transparent peer review scheme. This document only contains reviewer comments and rebuttal letters for versions considered at *Nature Communications*.

REVIEWER COMMENTS

Reviewer #1 (Remarks to the Author):

The authors have satisfactorily addressed the concerns raised on the previous version of the manuscript reviewed at a sister journal. So, I recommend publication in *Nature Communications*.

Reviewer #2 (Remarks to the Author):

Su et al. studied the structural mechanism of different ligands on b1AR-Gs coupling by analyzing cryoEM structures, MD simulation studies, cellular signaling with mutations, and BODIPY-GTP γ S uptake. They tried to understand the relationship between the functional outcome and the different structures caused by different ligands. CryoEM structures were almost identical among the b1AR-Gs complex structures with different ligands only with small local conformational differences at the ligand-binding residues and Gs-contacting residues. Although they showed that these residues are indeed involved in the b1AR-Gs coupling with mutation studies, they failed to connect the different ligand-binding structures with different Gs-interaction structures. For example, they did not show what structural differences T203 binding to cyanopindolol makes on the Gs-interacting residues. This kind of structural relationship between ligand-binding residues and Gs-interacting residues should be addressed for publication.

There are other issues.

1. Page 4, “since most full agonist” should be revised as the authors also suggested that there are structures with partial agonists.
2. Please use Ballesterio-Weinstein numbering system in figures, too.
3. In Fig 3a, it seems that V326A also affected Iso-induced cAMP production, however, the authors claimed that it only affected Dobutamine-induced cAMP production (page 7, first line). Please correct this if wrong and discuss the potential mechanism.
4. Did the authors observe local differences from Cryo-EM structures? They mainly discussed the local difference from MD simulation studies.
5. There are many other residues contacting Gs, please provide a rationale for choosing the mutation sites in Figure 5.

6. There are a few studies that made a mutation on the conserved large hydrophobic residue on ICL2 (residue F147A), and these studies showed that this mutation almost completely diminish the GPCR-Gs coupling. Please compare and discuss these papers with the current F147A data.

Reviewer #3 (Remarks to the Author):

In the revised manuscript, the authors addressed many of the major concerns that I had raised in the previous review report. Removal of the kinetic argument decreased an impact of the study, however, the study (with the new MD data) still provides an interesting thermodynamics insight into structural basis of ligand efficacy. There are, however, a couple of issues to be settled down before supporting publication of the study.

1) BW numbering

BW numbers should be labeled in the figure panels and the tables as well. This applies to Fig.2, 3, 4, 5, 6, 7f-g (y-axis), ExData Fig.7 (will it be better swap residue number and amino acid name?), 8, 9, 10, 13, 14 (y-axis).

2) IRA/RAi calculation (ExData Fig9a,b)

I recommend using Log-transformed values plus SEM to represent the IRA/RAi score because, as in the case of EC50 values, it shows a gaussian distribution in a logarithm scale.

3) Ligand titration for the kinetics analysis

The authors seems to be misunderstand my previous comment. I requested that ligand titration be performed at least for WT. The data will be plotted concentration versus the activation-phase parameter as well as the termination-phase parameters. EC50 values in this calculation are likely different from those in the 10-min endpoint cAMP assay (Fig.1a, Fig.3). If EC50 is right-shifted as compared with the endpoint cAMP data, interpretation of the mutant kinetics will require additional concern. For example, if pEC50 of Vo_ISO is 7.65, Vo_ISO analysis at 10 nM (Log -8) is mainly affected by ligand affinity as oppose to a change in signaling property.

Reviewer #5 (Remarks to the Author):

In this work, Su et al., obtained two cryo-EM structures of the beta 1 adrenergic receptor bound to Gs with a partial agonist - dobutamine or partial agonist - cyanopindolol. They have then used mutagenesis, MD simulations, and the BODIPY-GTPgammaS assay to further explore ligand efficacy at the structural level. They also compare the results on two ligands with the previously published by authors the cryo-EM complex of the beta 1 with a full agonist – isoproterenol. Overall, the conclusion they have is that the structures of the complexes are similar but there are local differences in the binding pocket. They also show different modulation of Gs by ligands. In the end, based on the BODIPY-GTPgammaS assay they show that there is different stability of the ligand-bound beta1-Gs complex in the nucleotide-free state. One can expect that there is a difference in different levels, and this has been shown before for several receptors. For some receptors, more details have been provided and key interactions have been mentioned. Please see Chris Tate's papers on the x-ray complex of beta 1 receptor with full and partial agonists. Although the manuscript has been improved taking into consideration the previous reviewer comments, the authors still clearly didn't show the mechanisms of differences and specific interactions that drive different efficacy of ligands. What kind of interactions stabilizes the binding of a full agonist, partial agonist, or weak agonist? How do these interactions communicate with the GPCR-Gs interface? I don't think that the manuscript has value for a broader audience and should be published in a more specified journal.

List of Manuscript Changes

We thank the reviewers very much for the helpful comments on our manuscript.

Reviewer #1:

“The authors have satisfactorily addressed the concerns raised on the previous version of the manuscript reviewed at a sister journal. So, I recommend publication in Nature Communications.”

We thank this reviewer for the time, effort and support!

List of Manuscript Changes

We thank the reviewers very much for the helpful comments on our manuscript.

Reviewer #2:

“Su et al. studied the structural mechanism of different ligands on b1AR-Gs coupling by analyzing cryoEM structures, MD simulation studies, cellular signaling with mutations, and BODIPY-GTPγS uptake. They tried to understand the relationship between the functional outcome and the different structures caused by different ligands. CryoEM structures were almost identical among the b1AR-Gs complex structures with different ligands only with small local conformational differences at the ligand-binding residues and Gs-contacting residues. Although they showed that these residues are indeed involved in the b1AR-Gs coupling with mutation studies, they failed to connect the different ligand-binding structures with different Gs-interaction structures. For example, they did not show what structural differences T203 binding to cyanopindolol makes on the Gs-interacting residues. This kind of structural relationship between ligand-binding residues and Gs-interacting residues should be addressed for publication.”

We thank this reviewer for the helpful comments. As this reviewer knows, the activation mechanism of class A GPCRs is conserved. The link between the ligand-binding site and Gs-interacting region is common including the CWxP, DRY, NPxxY and PIF motifs as shown in our Extended Data Figure 10. Our conclusion is that different ligands, together with G-proteins, lead to different stabilities of the entire ligand–receptor–G-protein complexes. We didn’t state that different ligand-binding structures would lead to different Gs-interacting structures.

1. *“There are other issues. Page 4, “since most full agonist” should be revised as the authors also suggested that there are structures with partial agonists.”*

As suggested, we have deleted this phrase.

2. *“Please use Ballesterro-Weinstein numbering system in figures, too.”*

As suggested, we have added the BW numbering in figures.

3. *“In Fig 3a, it seems that V326A also affected Iso-induced cAMP production, however, the authors claimed that it only affected Dobutamine-induced cAMP production (page 7, first line). Please correct this if wrong and discuss the potential mechanism.”*

As indicated in Figure 3b, the effect of V326A on Iso-induced cAMP production did not reach statistical significance.

4. *“Did the authors observe local differences from Cryo-EM structures? They mainly discussed the local difference from MD simulation studies.”*

As shown by the three-dimensional variability analysis (Supplementary Movies 1-3), the three complexes are flexible, same as seen in MD simulations.

5. *“There are many other residues contacting Gs, please provide a rationale for choosing the mutation sites in Figure 5.”*

As suggested, we have added one sentence: “We selected representative residues from TM5, TM6 and ICL2 since these regions contribute most to the interactions”.

6. *“There are a few studies that made a mutation on the conserved large hydrophobic residue on ICL2 (residue F147A), and these studies showed that this mutation almost completely diminish the GPCR-Gs coupling. Please compare and discuss these papers with the current F147A data.”*

As shown in Figure 5, F147A decreased the cAMP response by 63% in response to isoproterenol, and by 83% in response to dobutamine. The corresponding mutation (F139A) in β_2 -AR decreased the cAMP response by ~75% in response to isoproterenol (PMID: 8226735) which was in the same range as our data.

List of Manuscript Changes

We thank the reviewers very much for the helpful comments on our manuscript.

Reviewer #3:

“In the revised manuscript, the authors addressed many of the major concerns that I had raised in the previous review report. Removal of the kinetic argument decreased an impact of the study, however, the study (with the new MD data) still provides an interesting thermodynamics insight into structural basis of ligand efficacy. There are, however, a couple of issues to be settled down before supporting publication of the study.”

We thank this reviewer for the helpful comments.

1. *“BWnumbering. BW numbers should be labeled in the figure panels and the tables as well. This applies to Fig.2, 3, 4, 5, 6, 7f-g (y-axis), ExData Fig.7 (will it be better swap residue number and amino acid name?), 8, 9, 10, 13, 14 (y-axis).”*

As suggested, we have added the BW numbering in the figures and tables.

2. *“IRA/RAi calculation (ExData Fig9a,b). I recommend using Log-transformed values plus SEM to represent the IRA/RAi score because, as in the case of EC50 values, it shows a gaussian distribution in a logarithm scale.”*

As suggested, we have used LogIRA.

3. *“Ligand titration for the kinetics analysis. The authors seems to be misunderstand my previous comment. I requested that ligand titration be performed at least for WT. The data will be plotted concentration versus the activation-phase parameter as well as the termination-phase parameters. EC50 values in this calculation are likely different from those in the 10-min endpoint cAMP assay (Fig.1a, Fig.3). If EC50 is right-shifted as compared with the endpoint cAMP data, interpretation of the mutant kinetics will require additional concern. For example, if pEC50 of Vo_ISO is 7.65, Vo_ISO analysis at 10 nM (Log -8) is mainly affected by ligand affinity as oppose to a change in signaling property.”*

Figure 6 shows that mutations of the G-protein interacting residues are also affecting the kinetics of the cAMP responses induced by the different ligands. We used EC90 (not EC50) concentrations for individual ligands. These mutated residues were in the receptor/G-protein interface, not in the ligand-binding site. Following this reviewer’s suggestion, we have performed the control ligand titration experiments for wild-type β_1 -AR (Please see the image below). After plotting the curves of concentrations versus the activation-phase parameter as well as the

termination-phase parameter, EC50 values were similar to the EC50 values in Figures 3, 5, 6, and Extended Data Figure 9.

List of Manuscript Changes

We thank the reviewers very much for the helpful comments on our manuscript.

Reviewer #5:

“In this work, Su et al., obtained two cryo-EM structures of the beta 1 adrenergic receptor bound to Gs with a partial agonist - dobutamine or partial agonist - cyanopindolol. They have then used mutagenesis, MD simulations, and the BODIPY-GTPgammaS assay to further explore ligand efficacy at the structural level. They also compare the results on two ligands with the previously published by authors the cryo-EM complex of the beta 1 with a full agonist – isoproterenol. Overall, the conclusion they have is that the structures of the complexes are similar but there are local differences in the binding pocket. They also show different modulation of Gs by ligands. In the end, based on the BODIPY-GTPgammaS assay they show that there is different stability of the ligand-bound beta1-Gs complex in the nucleotide-free state. One can expect that there is a difference in different levels, and this has been shown before for several receptors. For some receptors, more details have been provided and key interactions have been mentioned. Please see Chris Tate’s papers on the x-ray complex of beta 1 receptor with full and partial agonists. Although the manuscript has been improved taking into consideration the previous reviewer comments, the authors still clearly didn’t show the mechanisms of differences and specific interactions that drive different efficacy of ligands. What kind of interactions stabilizes the binding of a full agonist, partial agonist, or weak agonist? How do these interactions communicate with the GPCR-Gs interface? I don’t think that the manuscript has value for a broader audience and should be published in a more specified journal.”

In this manuscript, we have investigated the ligand efficacy by examining the entire complex (the ligand–receptor–G-protein complex). We conclude that the ligand efficacy correlates with the stability of the entire complex. We have identified the specific residues on the receptor that are involved in the specific interactions with the different ligands and with G-proteins (Figures 2, 3, 4 and 5, Extended Data Figures 3, 7 and 8). Regarding the connection between the ligand-binding site and the G-protein interface, the activation mechanism for class A GPCRs is conserved. We have discussed in the manuscript (Extended Data Figure 10). There are no different communication pathways from the ligand-binding site to the Gs interface for ligands with different efficacies.

REVIEWERS' COMMENTS

Reviewer #2 (Remarks to the Author):

The main concern about the manuscript was that the cryo-EM structures of Isoproterenol-bound, Dobutamine-bound, and Cyanopindolol-bound states do not suggest the mechanism of different agonist-induced different G protein activation.

1. The authors claimed, "All three b1-AR structures display similar conformational changes of these residues; no intermediate conformations are observed in the presence of partial agonists (Extended Data Fig 10b)." and "Therefore, b1-ARs in the three different complexes, with Gs-proteins, adopt similar active state conformations, even though they are bound with ligands with different efficacies."

However, with MD simulation studies, the authors also claimed, "Compared with the isoproterenol-b1-AR-Gs, the dobutamine-b1-AR-Gs structure showed different flexibilities in the ligand-binding pocket, TM1, ICL1, TM2, ICL2, TM4, ECL3, TM7 and H8 of b1-AR, as well as local regions of Gas and Gbg (Fig. 4b). The cyanopindolol-b1-AR-Gs complex also showed different flexibilities in the ligand-binding pocket, TM1, ICL1, ICL2, TM5, TM6, TM7 and H8 of b1-AR, as well as local regions of Gas and Gbg (Fig. 4c). Overall, the cyanopindolol-b1-AR-Gs complex is relatively less flexible (Fig. 4c). The residues in b1-AR that are involved in Gs interactions showed varied flexibilities in the three complexes (Fig. 4 d-f)." and "Overall, the GaMD simulations support our cryo-EM structural data showing local conformational differences among the three complexes."

The authors have not explained how the different ligand-binding shown in the cryo-EM structures induce different flexibilities or how these different flexibilities are relevant to the different G protein activation.

2. In addition, the authors have tested different mutations on the different ligand-induced G protein activation (Figs 5 and 6). However, they did not explain how the different ligand-binding shown in the cryo-EM structures induces these different mutant effects.

3. As a last data, the authors performed GTPγS incorporation kinetics analysis and related MD simulation (Fig 7). The results are "The isoproterenol-b1-AR-Gs complex has the longest half-life and is thus the most stable among the three complexes (Fig. 7e). The cyanopindolol-b1-AR-Gs complex is the least stable, displaying the shortest half-life (Fig. 7e). The dobutamine-b1-AR-Gs complex displays an intermediate stability (Fig. 7e)." and "These GaMD simulations thus reveal that ligands with higher efficacies are able to maintain b1-AR in the active state for longer time. These data are consistent with our above biochemical data."

Again, with these data, the authors did not explain how the different ligand-binding shown in the cryo-EM structures induces this different G protein activation kinetics.

Minor corrections

In the first subtype of Results, please consistent with using capital letters.

Reviewer #3 (Remarks to the Author):

In the revised manuscript, the authors adequately addressed most of my previous points.

I strongly recommend including the ligand-titration data in Ex Data Fig13 because, as I pointed previously, apparent potency may change depending on responses to be measured (endpoint cAMP level vs kinetics) and it is uncertain unless experimental data are presented.

In relation to this, the variables of the parameters (SD) looked oddly tighter than usual cell-based experiments. Do the data represent merged results from three independent experiments as indicated in the figure legend "Data are presented as mean \pm SD (n=3)"? I note that 'n' stands for independent experiments, not technical replicate.

List of Manuscript Changes

We thank the reviewers very much for the helpful comments on our manuscript.

Reviewer #2:

1. *“The main concern about the manuscript was that the cryo-EM structures of Isoproterenol-bound, Dobutamine-bound, and Cyanopindolol-bound states do not suggest the mechanism of different agonist-induced different G protein activation.*

1. *The authors claimed, “All three b1-AR structures display similar conformational changes of these residues; no intermediate conformations are observed in the presence of partial agonists (Extended Data Fig 10b).” and “Therefore, b1-ARs in the three different complexes, with Gs-proteins, adopt similar active state conformations, even though they are bound with ligands with different efficacies.”*

However, with MD simulation studies, the authors also claimed, “Compared with the isoproterenol–b1-AR–Gs, the dobutamine–b1-AR–Gs structure showed different flexibilities in the ligand-binding pocket, TM1, ICL1, TM2, ICL2, TM4, ECL3, TM7 and H8 of b1-AR, as well as local regions of Gas and Gbg (Fig. 4b). The cyanopindolol–b1-AR–Gs complex also showed different flexibilities in the ligand-binding pocket, TM1, ICL1, ICL2, TM5, TM6, TM7 and H8 of b1-AR, as well as local regions of Gas and Gbg (Fig. 4c). Overall, the cyanopindolol–b1-AR–Gs complex is relatively less flexible (Fig. 4c). The residues in b1-AR that are involved in Gs interactions showed varied flexibilities in the three complexes (Fig. 4 d-f).” and “Overall, the GaMD simulations support our cryo-EM structural data showing local conformational differences among the three complexes.”

The authors have not explained how the different ligand-binding shown in the cryo-EM structures induce different flexibilities or how these different flexibilities are relevant to the different G protein activation.”

We thank this reviewer for the helpful comments. The main conclusion of this paper is that different signaling complexes (the entire ligand–receptor–G-protein complexes) have similar overall complex architecture, with local conformational differences. These lead to different overall stabilities. The more stable of a complex, the ligand-receptor stays in an active conformation longer, and thus activates more G-proteins (a higher efficacy).

2. *“In addition, the authors have tested different mutations on the different ligand-induced G protein activation (Figs 5 and 6). However, they did not explain how the different ligand-binding shown in the cryo-EM structures induces these different mutant effects.”*

If this reviewer was asking for a specific pathway(s) from a specific structural difference in the ligand-binding pocket to a specific change in the G-protein interacting site, we did not investigate here.

3. *“As a last data, the authors performed GTPγS incorporation kinetics analysis and related MD simulation (Fig 7). The results are “The isoproterenol–b1-AR–Gs complex has the longest half-*

life and is thus the most stable among the three complexes (Fig. 7e). The cyanopindolol–b1-AR–Gs complex is the least stable, displaying the shortest half-life (Fig. 7e). The dobutamine–b1-AR–Gs complex displays an intermediate stability (Fig. 7e).” and “These GaMD simulations thus reveal that ligands with higher efficacies are able to maintain b1-AR in the active state for longer time. These data are consistent with our above biochemical data.” Again, with these data, the authors did not explain how the different ligand-binding shown in the cryo-EM structures induces this different G protein activation kinetics.”

The main point of the paper is that receptors bound with ligands of different efficacies have similar overall conformations, and the efficacy correlates with the stability of the complex. We did not investigate how the different ligand-bindings specifically induce the different G-protein activation kinetics.

“Minor corrections

In the first subtype of Results, please consistent with using capital letters.”

As suggested, we have changed to capital letters.

List of Manuscript Changes

We thank the reviewers very much for the helpful comments on our manuscript.

Reviewer #3:

“In the revised manuscript, the authors adequately addressed most of my previous points. I strongly recommend including the ligand-titration data in Ex Data Fig13 because, as I pointed previously, apparent potency may change depending on responses to be measured (endpoint cAMP level vs kinetics) and it is uncertain unless experimental data are presented. In relation to this, the variables of the parameters (SD) looked oddly tighter than usual cell-based experiments. Do the data represent merged results from three independent experiments as indicated in the figure legend "Data are presented as mean \pm SD (n=3)"? I note that 'n' stands for independent experiments, not technical replicate.”

As suggested, we have included the ligand-titration data in Supplementary Fig. 13. The data represents three independent experiments.